# Domain-invariant Learning using Adaptive Filter Decomposition

## Abstract

Domain shifts are frequently encountered in real-world scenarios. In this paper, we consider the problem of domain-invariant deep learning by explicitly modeling domain shifts with only a small amount of domain-specific parameters in a Convolutional Neural Network (CNN). By exploiting the observation that a convolutional filter can be well approximated as a linear combination of a small set of basis elements, we show for the first time, both empirically and theoretically, that domain shifts can be effectively handled by decomposing a regular convolutional layer into a domain-specific basis layer and a domain-shared basis coefficient layer, while both remain convolutional. An input channel will now first convolve spatially only with each respective domain-specific basis to "absorb" domain variations, and then output channels are linearly combined using common basis coefficients trained to promote shared semantics across domains. We use toy examples, rigorous analysis, and real-world examples to show the framework's effectiveness in cross-domain performance and domain adaptation. With the proposed architecture, we need only a small set of basis elements to model each additional domain, which brings a negligible amount of additional parameters, typically a few hundred.

## 1 Introduction

Training supervised deep networks requires large amount of labeled training data; however, well-trained deep networks often degrade dramatically on testing data from a significantly different domain. In real-world scenarios, such domain shifts are introduced by many factors, such as different illumination, viewing angles, and resolutions. Research topics such as transfer learning and domain adaptation are studied to promote invariant representations across domains with different levels of availabilities of annotated data.

Recent efforts on learning cross-domain invariant representations using deep networks generally fall into two categories. The first one is to learn a common network with constraints encouraging invariant feature representations across different domains Long et al. (2015; 2016b); Tzeng et al. (2014). The feature invariance is usually measured by feature statistics like Maximum Mean Discrepancy (MMD), or feature discriminators using adversarial training Ganin et al. (2016). While these methods introduce no additional model parameters, the effectiveness largely depends on the degree of domain shifts. The other direction is to explicitly model domain specific characteristics with a multi-stream network structure where different domains are modeled by corresponding sub-networks at the cost of significant extra parameters and computations Rozantsev et al. (2018b). Regularizations are imposed among sub-networks to encourage common semantics across domains.

In this paper, we model domain shifts through domain-adaptive filter decomposition (DAFD) with layer branching. At a branched layer, we decompose each filter over a small set of domain-specific basis elements to model intrinsic domain characteristics, while enforcing shared cross-domain coefficients to align invariant semantics. A regular convolution is now decomposed into two steps. First, a domain-specific basis convolves spatially only each individual input channel for shift "correction." Second, the "corrected" output channels are weighted summed using domain-shared basis coefficients ($1 \times 1$ convolution) to promote common semantics. When domain shifts happen in space, we rigorously prove that such layer-wise "correction" by the same spatial transform applied to bases suffices to align the learned features.

Comparing to the existing subnetwork-based methods, the proposed method has several appealing properties: First, only a very small amount of additional trainable parameters are introduced to explicitly model each domain, i.e., domain-specific bases. The majority of the parameters in the

network remain shared across domains, and learned from abundant training data to effectively avoid overfitting. Furthermore, the decomposed filters reduce the overall computations significantly compared to previous works, where computation typically grows linearly with the number of domains.

We conduct extensive real-world face recognition, image classification, and segmentation experiments, and observe that, with the proposed method, invariant representations across domains are consistently achieved without compromising the performance of individual domain.

Our main contributions are summarized as follows:

- We propose domain-invariant representation learning through bases decomposition with layer branching, where domain-specific bases are learned to counter domain shifts, and semantic alignments are enforced with cross-domain common basis coefficients.
- We both theoretically prove and empirically observe that by stacking the bases-decomposed branched layer, invariant representations across domains are achieved progressively.
- The majority of network parameters remain shared across domains, which alleviates the demand for massive annotated data from every domain, and introduces only a small amount of additional computation and parameter overhead. Thus the proposed approach serves as an efficient way for domain invariant learning.

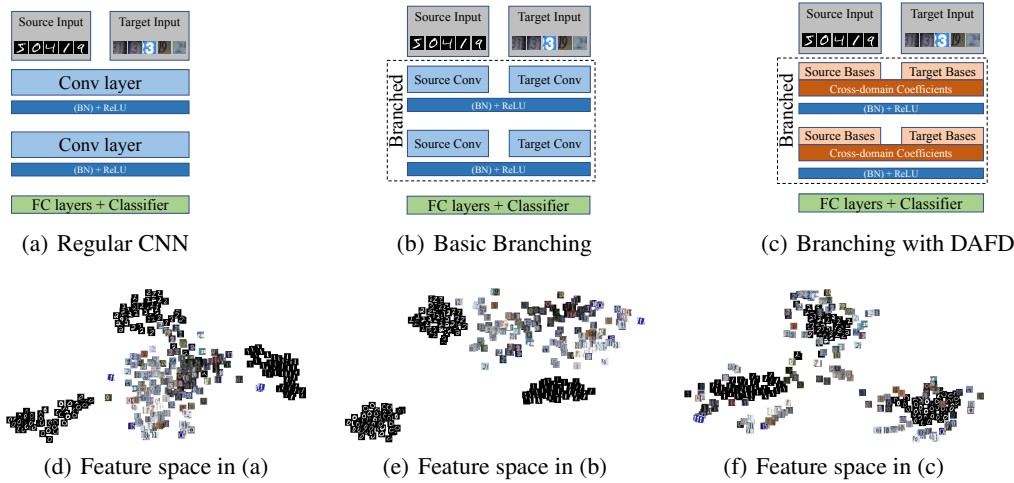

(a) Regular CNN  (b) Basic Branching  (c) Branching with DAFD

(d) Feature space in (a)  (e) Feature space in (b)  (f) Feature space in (c)

Figure 1: Three candidate architectures considered for domain-invariant representation learning. In (a), a set of common network parameters are trained to model both source and target domains. In (b), the domain characteristics are explicitly modeled by two sets of convolutional filters in each convolutional layer. Our approach is illustrated in (c) where domain-adaptive bases are learned to "absorb" domain shifts, while the decomposition coefficients are shared across domains to promote common semantics. The feature space of the three candidate architectures, MNIST → SVHN, are visualized using t-SNE Maaten & Hinton (2008) in (d), (e), (f), respectively. The obtained cross-domain invariance can be clearly observed in (f).

## 2 RELATED WORK

Recent achievements on domain-invariant learning generally follow two directions. The first direction is learning a single network, where parameters are shared across domains, while the network is encouraged to produce domain-invariant features by minimizing additional loss functions in the network training Ganin et al. (2016); Long et al. (2015; 2016a;b); Tzeng et al. (2014). The Maximum Mean Discrepancy (MMD) Gretton et al. (2007), and MK-MMD Gretton et al. (2012) in Long et al. (2015), are adopted as the discrepancy metric among domains. Beyond the first order statistic, second-order statistics are utilized in Hoffman et al. (2014). Besides the hand-crafted distribution distance metrics, Ganin et al. (2016); Tzeng et al. (2015); Long et al. (2018) resort to adversarial training and achieve superior performances. Various distribution alignment methods, e.g., Wu et al. (2019); Kumar et al. (2018), are proposed to improve the invariant feature learning. While effective in certain scenarios, the performance of learning invariant features using a shared network is largely constrained by the degree of domain shift as discussed in Rozantsev et al. (2018b). Meanwhile,

some recent works like Wu et al. (2019); Zhao et al. (2019) suggest important insights on whether it is sufficient to do domain adaptation by invariant representation and small empirical source risk, which shed light on exploring more effective alignment methods that are robust to common issues like different marginal label distributions.

Another popular direction is modeling each domain explicitly using auxiliary network structures. Based on the hypothesis that explicitly modeling what is unique to each domain can improve a model's ability to extract domain-invariant features, e.g., Bousmalis et al. (2016) proposes feature representation by two components where domain similarities and shifts are modeled by a private component and a shared component separately. A completely two-stream network structure with no shared parameters is proposed in Rozantsev et al. (2018b), where auxiliary residual networks are trained to adapt the layer parameters of the source domain to the target domain. Chang et al. (2019) proposes attacking domain shifts by domain-specific batch normalization, which we believe is compatible with the proposed domain-specific filter decomposition for better performance. Another popular direction for domain adaptation is to remap the input data between the source and the target domain for domain adaptation Murez et al. (2017); Hu et al. (2018); Hoffman et al. (2018), which is not included in the discussion since we are focusing on learning invariant feature space. Finally, learning invariance is of relevance beyond domain adaptation, e.g., in the field of causal inference Bühlmann (2018).

## 3 Domain-adaptive Filter Decomposition

A straightforward way to address domain shifts is to learn from multi-domain training data a single network as in Figure 1(a). However, the lack of explicitly modelling of individual domains often results in unnecessary information loss and performance degradation as discussed in Rozantsev et al. (2018b). Thus, we often simultaneously observe underfitting for domains with abundant training data, and overfitting for domains with limited training. In this section, we start with a simplistic pedagogical formulation, domain-adaptive layer branching as in Figure 1(b), where domain shifts are modeled by a respective branch of filters in a layer, one branch per domain. Each branch is learned only from domain-specific data, while non-branched layers are learned from data from all domains. We then propose to extend basic branching to bases-decomposed branching as in Figure 1(c), where domain characteristics are modeled by domain-specific bases, and shared decomposition coefficients are enforced to align cross-domain semantics.

### 3.1 Pedagogical Branching Formulation

We start with the simple-minded branching formulation in Figure 1(b). To model the domain-specific characteristics, at the first several convolutional layers, we dedicate a separate branch to each domain. Domain shifts are modeled by an independent set of convolutional filters in the branch, trained respectively with errors propagated back from the loss functions of source and target domains. For supervised learning, the loss function is the cross-entropy for each domain. For unsupervised learning, the loss function for the target domain can be either the feature statistics loss or the adversarial loss. The remaining layers are shared across domains. We assume one target domain and one source domain in our discussion, while multiple domains are supported. Note that, though we adopt the source vs. target naming convention in the domain adaptation literature, we address here a general domain-invariant learning problem.

Domain-adaptive branching is simple and straightforward, however, it has the following drawbacks: First, both the number of model parameters and computation are multiplied with the number of domains. Second, with limited target domain training data, we can experience overfitting in determining a large amount of parameters dedicated to that domain. Third, no constraints are enforced to encourage cross-domain shared semantics. We address these issues through layer branching with the proposed bases decomposition.

### 3.2 Bases-decomposed Branching

To simultaneously counter domain shifts and enforce cross-domain shared semantics, we decompose each convolutional filter in a branched layer into domain-specific bases, and cross-domain shared coefficients, as illustrated in Figure 1(c).

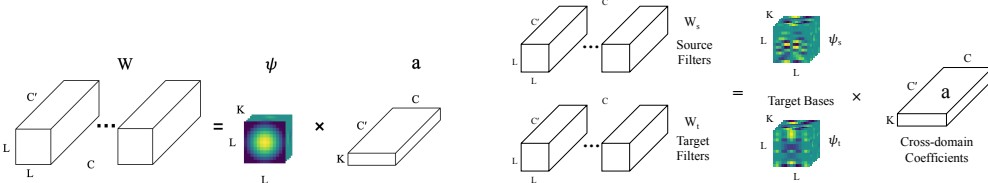

(a) Decomposition over pre-fixed bases Qiu et al. (2018).

(b) The proposed domain-adaptive filter decomposition.

Figure 2: Illustrations of the proposed domain-adaptive filter decomposition.

**Filter Decomposition.** A convolutional filter can be well approximated as a linear combination of a small set of basis elements Qiu et al. (2018). In Figure 2(a), a convolutional filter $W$ with a size of $L \times L \times C' \times C$ is decomposed into a truncated expansion with pre-fixed bases in the spatial domain. The truncated bases $\psi$ with a size $L \times L \times K$ are predefined and remain fixed in training, where $K$ is a hyperparameter, usually a small number, e.g., $K = 5$, indicating the number of basis elements used in each layer. The trainable parameters are thus reduced to the decomposition coefficients $a$ with a size of $K \times C' \times C$. With bases decomposition, a reduction rate of $\frac{K}{L^2}$ is achieved in both model complexity and computation. This parameter saving is not the only virtue of basis decomposition, such decomposition is natural to extract both invariance across domains and adaptation to specific domains as detailed next.

In our approach, as shown in Figure 2(b), we decompose source and target domain filters over domain-specific bases, with decomposition coefficients shared across domains. Specifically, at each branched layer, the source domain filter $W_s$ and target domain filter $W_t$ of size $L \times L \times C' \times C$, are decomposed into $\psi_s \times a$ and $\psi_t \times a$, where $\psi_s$ and $\psi_t$ with a size of $L \times L \times K$ are the domain-adaptive bases for source and target domains, respectively; and $a \in \mathbb{R}^{K \times C' \times C}$ denotes the common decomposed coefficients shared across domains. Different from Qiu et al. (2018), the domain-adaptive bases are independently learned from the corresponding domain data to model domain shifts, and the shared decomposed coefficients are learned from the massive data from multiple domains. Note that thanks to this proposed structure, only a small amount of additional parameters is required here to model each additional domain, typically a few hundred.

With the above domain-adaptive filter decomposition, at each branched layer, a regular convolution is now decomposed into two: First, a domain-specific basis convolves each individual input channel for domain shift "correction." Second, the "corrected" output channels are weighted summed using domain-shared basis coefficients (1×1 convolution) to promote common semantics. A toy example is presented in Figure A.1 for illustrating the intuition behind the reason why manipulating bases alone can address domain shifts. We generate target domain data by applying two trivial operations to source domain images: First, every $3 \times 3$ non-overlapping patch in each image is locally rotated by $90^o$. Then, images are negated by multiplying with $-1$. Domain-invariant features are observed by manipulating bases alone. We will rigorously prove in Section 4 why such layer-wise "correction" aligns features across domains, and present real-world examples in the experiments.

**Parameters and Computation Reduction.** Suppose that both input and output features have the same spatial resolution of $W \times W$, in each forward pass in a regular convolutional layer, there are totally $W^2 \times C' \times C \times (2L^2 + 1)$ flops for each domain. While in our model, each domain only introduces $W^2 \times C' \times 2K(L^2 + C)$ flops, where $K$ is the number of basis elements. For parameters, there are totally $D \times C' \times C \times L^2$ parameters in a regular convolutional layer where $D$ is the number of domains which is typically 2 in our case. In our model, each layer has only $K \times (C' \times C + D \times L^2)$ parameters. Taking VGG-16 Simonyan & Zisserman (2014) as an example with an input size of $224 \times 224$, a regular VGG-16 with branching, Fig 1(b) and Rozantsev et al. (2018a;b), requires adding 14.71M parameters and 15.38G flops in convolutional layers to handle each additional domain. With the here proposed method (Fig 1(c)), a VGG-16 only requires adding 702 parameters and 10.75G flops to handle one additional domain (K=6).

## 4 PROVABLE INVARIANCE WITH ADAPTIVE BASES

In this section, we theoretically prove that the features produced by the source and target networks from domain-transferred inputs can be aligned by the proposed framework of only adjusting multi-

layer bases, assuming a generative model of the source and target domain images via CNN. Since convolutional generative networks are a rich class of models for domain transfer Hu et al. (2018); Murez et al. (2017), our analysis provides a theoretical justification of the proposed approach. Diverse examples in the experiment section shows the applicability of the proposed approach is potentially larger than what is proved here. All proofs are in the supplementary material.

**Filter Transform via Basis Transform.** Let $w_s$ and $w_t$ be the filters in the branched convolutional layer for the source and target domains respectively, and similarly denote the source and target bases by $\psi_{k,s}$ and $\psi_{k,t}$. In the proposed bases decomposition DA architecture, the source and target domain filters are linear combinations of the domain-specific basis elements with shared decomposition coefficients, namely

$$w_s(u) = \sum_k a_k \psi_{k,s}(u), \quad w_t(u) = \sum_k a_k \psi_{k,t}(u).$$

Certain transforms of the filter can be implemented by only transforming the bases, including

(1) A linear correspondence of filter values. Let $\lambda : \mathbb{R} \to \mathbb{R}$ be a linear mapping, by linearity,

$$\psi_{k,s}(u) \to \psi_{k,t}(u) = \lambda(\psi_{k,s}(u)) \qquad \text{applies } w_s(u) \to w_t(u) = \lambda(w_s(u)).$$

E.g. the negation $\lambda(\xi) = -\lambda(\xi)$, as shown in Figure A.1.

(2) The transformation induced by a displacement of spatial variable, i.e., "spatial transform" of filters, defined as $D_\tau w(u) = w(u - \tau(u))$, where $\tau : \mathbb{R}^2 \to \mathbb{R}^2$ is a differentiable displacement field. Note that the dependence on spatial variable $u$ in a filter is via the bases, thus $\psi_{k,s} \to \psi_{k,t} = D_\tau \psi_{k,s}$ applies $w_s \to w_t = D_\tau w_s$.

If such filter adaptations are desired in the branching network, then it suffices to branch the bases while keeping the coefficients $a_k$ shared, as implemented in the proposed architecture shown in Figure 1(c). A fundamental question is thus how large is the class of possible domain shifts that can be corrected by these "allowable" filter transforms. In the rest of the section, we show that if the domain shifts in the images are induced from a generative CNN where the filters for source and target differ by a sequence of allowable transforms, then the domain shift can be provably eliminated by another sequence of filter transforms which can be implemented by bases branching only.

**Correction of a Single Filter Transform.** We first analyze the "symmetric" correction of one filter spatial transform $D_\tau$ in one layer. The inclusion of linear correspondence transform is more direct (see comments after Theorem 1). For technical reasons, we assume that the displacement field $\tau$ is a small distortion, namely $\|\nabla \tau\|_\infty \ll 1$, and then $D_\tau$ is invertible. Example includes rotation by a small angle and a small factor rescaling (dilation).

For simplicity we only consider one input and output channel in each of the multiple convolutional layers. The argument extends to multiple channels by modifying the boundedness condition of the filters. Then the forward mapping in one convolutional layer can be written as $y = \sigma(x * w + b)$, where $x$ is the input activation, $y$ is the output, $w$ is the filter, $b$ is the constant bias, and $\sigma$ is the nonlinear activation function, e.g., ReLU. As we take a continuous formulation in the analysis, the activations $x$ and $y$ are assumed to be smooth functions supported on domain $\Omega \subset \mathbb{R}^2$, typically $\Omega = [-1,1]^2$. The filter $w$ is a function supported on $2^j B$, $B$ being the unit disk, and $2^j$ is layer scale (diameter of filter patches). The 1-norm of a function is defined to be $\|x\|_1 = \int_{\mathbb{R}^2} |x(u)| du$.

**Lemma 1.** *Suppose that the two filters $w$, $f$ are supported on $2^{j_w} B$ and $2^{j_f} B$ respectively. $\sigma : \mathbb{R} \to \mathbb{R}$ is non-expansive, $D_\tau$ is a spatial transform where $\tau$ is odd, i.e., $\tau(-u) = -\tau(u)$, and $|\nabla \tau|_\infty < \frac{1}{5}$. Then*

$$\|\sigma_b(x * D_\tau w) * f - \sigma_b(x * w) * D_\tau^{-1} f\|_1$$
$$\leq 2|\nabla \tau|_\infty \|w\|_1 \|f\|_1 \left\{ (2^{j_w} + 2^{j_f}) \|\nabla x\|_1 + 4\|x\|_1 \right\},$$

*where $\sigma_b$ denotes the nonlinear function with the bias. The second term vanishes if $(I_d - \tau)$ is a rigid motion, e.g., rotation.*

**Provable Invariance under a Generative Model of Images.** Stacking the approximate commuting relation, Lemma 1, in multiple layers allows to correct a sequence of filter transforms in previous convolutional layers by another sequence of "symmetric" ones. This means that if we impose a convolutional generative model on the source and target input images, and assume that the domain transfer results from a sequence of spatial transforms of filters in the generative net, then by correcting these filter transforms in the subsequent convolutional layers we can guarantee the recovery of

the same feature mapping. The key observation is that the filter transfers can be implemented by bases transfer only.

We summarize the theoretical assumptions as follows:

(A1) The $\sigma$ in any layer is non-expansive,

(A2) In the generative net (where layer is indexed by negative integers), $w_t^{(-l)} = D_l w_s^{(-l)}$, where $D_l = D_{\tau_l}$, $\tau_l$ is odd and $|\nabla \tau_l|_\infty \leq \varepsilon < \frac{1}{5}$ for all $l = 1, \cdots, L$. The biases in the target generative net are mildly adjusted accordingly due to technical reasons (to preserve the "baseline output" from zero-input, c.f. the proof).

(A3) In the generative net, $\|w_s^{(-l)}\|_1 \leq 1$ for all $l$, and so is $w_t^{(-l)} = D_l w_s^{(-l)}$. Same for the feed-forward convolutional net taking the generated images as input, called "feature net": The source net filters have $\|w_s^{(l)}\|_1 \leq 1$ for $l = 1, 2 \cdots$, and same with $D_l w_s^{(l)}$ which will be set to be $w_t^{(l)}$. Also, $w_s^{(-l)}$ and $w_s^{(l)}$ are both supported on $2^{j_l} B$ for $l = 1, \cdots, L$.

One can show that $\|D_\tau w\|_1 = \|w\|_1$ when $(I_d - \rho)$ is a rigid motion, and generally $|\|D_\tau w\|_1 - \|w\|_1| \leq c|\nabla \tau|_\infty \|w\|_1$ which is negligible when $\varepsilon$ is small. Thus in (A3) the boundedness of the 1-norm of the source and target filters imply one another exactly or approximately. The boundedness of 1-norm of the filters preserves the non-expansiveness of the mapping from input to output in a convolutional layer, and in practice is qualitatively preserved by normalization layers. Also, as a typical setting, (A3) assumes that the scales $j_l$ in the generative net (the $(-l)$-th layer) and the feature net (the $l$-th layer) are matched, which simplifies the analysis and can be relaxed.

**Theorem 1.** *Suppose that $X_s$ and $X_t$ are source and target images generated by $L$-layer generative CNN nets with source and target filters $w_s^{(-l)}$, $w_t^{(-l)}$ respectively from the common representation $h$. Under (A1)-(A3), the output at the $L$-th layer of the target feature CNN from $X_t$, by setting $w_t^{(l)} = D_l w_s^{(l)}$ in all layers which can be implemented by bases branching, approximates that of the source feature CNN from $X_s$ up to an error which is bounded in 1-norm by*

$$4\varepsilon \left\{ (\sum_{l=1}^{L} 2^{j_l}) \|\nabla h\|_1 + 2L\|h\|_1 \right\},$$

*and the second term vanishes if $(I_d - \tau_l)$ are rigid motions, e.g., rotation.*

Note that typically $j_1 \leq \cdots \leq j_L$ and $2^{J_L} \leq 1$, and $\sum_l 2^{j_l}$ is proportional to the diameter of $\Omega$ and thus an $O(1)$ constant. The result can be extended to allow a linear transform of the filter values, namely $Dw(u) = \lambda(D_\tau w(u))$, and correct the target filters $f_t$ to be $\lambda^{-1}(D_\tau f_s)$. In Lemma 1, $\lambda$ commutes with the convolution by linearity, and also with $\sigma$ by adjusting the bias parameter if needed. Unlike $D_\tau$ which becomes $D_\tau^{-1}$ when applying to $f$, $\lambda$ remains as $\lambda$, thus $\lambda^{-1}$ is needed in the symmetric basis transfers to correct the domain shift. Proofs are in the supplementary material.

## 5 EXPERIMENTS

In this section, we perform extensive experiments to evaluate the performance of the proposed domain-adaptive filter decomposition. We start with the comparisons among the 3 architectures listed in Figure 1 on two supervised tasks. To demonstrate the proposed framework as one principled way for domain-invariant learning, we then conduct a set of domain adaptation experiments. There we show, by simply plugging the proposed domain filter decomposition into regular CNNs used in existing domain adaptation methods, we consistently observe performance improvements, which illustrates that the framework can be successfully applied with multiple leading deep network architectures and very different datasets and applications.

### 5.1 ARCHITECTURE COMPARISONS

We start with two supervised tasks performed on the three architectures listed in Figure 1, regular CNN (A1), basic branching (A2), and branching with domain-adaptive filter decomposition (A3). In each layer of the network with the proposed domain-adaptive filter decomposition, the different domain features convolve spatially with the corresponding domain-specific bases first, then the output features are linearly combined by the cross-domain basis coefficients simultaneously. For both tasks, the networks are trained end-to-end with a summed loss for domains, and the domain-specific bases are only updated by the error from the corresponding domain, while the basis coefficients are updated by the joint error across domains.

| Scales | Source domain | | | Target domain | | |
|---|---|---|---|---|---|---|
| | 0.1 | 0.05 | 0.005 | 0.1 | 0.05 | 0.005 |
| A1 | 98.4 | 96.4 | 98.0 | 81.6 | 80.2 | 61.0 |
| A2 | 99.2 | 98.6 | 97.6 | 81.4 | 78.4 | 49.6 |
| **A3** | **99.4** | **98.8** | **98.8** | **85.6** | **82.2** | **64.4** |

(a) Accuracy (%) on both domains for supervised domain adaptation.

| Methods | VIS Acc (%) | NIR Acc (%) | NIR+VIS (%) |
|---|---|---|---|
| A1 | 75.57 | 52.71 | 98.44 |
| A2 | 94.46 | 87.50 | 98.58 |
| **A3** | **97.16** | **95.03** | **99.15** |

(b) Cross-domain simultaneous face recognition on NIR-VIS-2.0.

Table 1: Comparisons on supervised domain adaptaion and cross-domain simultaneous face recognition. A1, A2, and A3 correspond to regular CNN, basic branching, and branching with domain-adaptive filter decomposition shown in Figure 1, respectively.

**Supervised Domain Adaptation on Digits.** The first task is supervised domain adaptation, where we adopt a challenging setting by using MNIST handwritten digit dataset as the source domain, and SVHN as the target domain. We perform a series of experiments by progressively reducing the annotated training data for the target domain. We start the comparisons at 10% of the target domain labeled samples, and end at 0.5% where only 366 labeled samples are available for the target domain. The results on test set for both domains are presented in Table 3(a). It is clearly shown that when training the target domain with small amount of data, a network with basic branching suffers from overfitting to the target domain because of the large amount of domain specific parameters. While regular CNN generates well on target domain, the performance on source domain degrades when the number of target domain data is comparable. A network with the proposed domain-adaptive filter decomposition significantly balances the learning of both the source and the target domain, and achieves best accuracies on both domains regardless of the amount of annotated training data for the target domain. The feature space of the three candidate architectures are visualized in Figure 1.

**Supervised Simultaneous Cross-domain Face Recognition.** To show the generality of the proposed domain-adaptive filter decomposition for various tasks, we further perform a supervised cross-domain (RGB and near infrared) *simultaneous* face recognition experiment in the supplementary material Section B, and the quantitative comparisons are in Table 3(b), where we can clearly observe that DAFD performs superiorly even with a missing input domain. Note that A2 requires additional 14.71M parameters over A1, while our method requires only 0.0007M as shown in Table A.1.

## 5.2 EXPERIMENTS ON STANDARD DOMAIN ADAPTATIONS

In this section, we perform extensive experiments on unsupervised domain adaptation. Note that the objective of the experiments in this section is not to validate the proposed domain-adaptive filter decomposition as just another new method for domain adaptation. Instead, since most of the state-of-the-art domain adaptation methods adopt the regular CNN (A1) with completely shared parameters for domains, we show the compatibility and the generality of the proposed domain-adapting filter decomposition by plugging it into underlying domain adaptation methods, and evaluate the effectiveness by retraining the networks using exactly the same setting and observing the performance improvement over the underlying methods. Diverse real-world domain shifts including different sensors, different image sources, and synthetic images, and applications on both classification and segmentation are examined in these experiments. Together with the experiments in the previous section, this further stresses the plug-and-play virtue of the proposed framework.

In practise, instead of learning independent source and target domain bases, we learn the *residual* between the source and the target domain bases. The residual is initialized by full zeros, and trained by loss for encouraging invariant features in the underlying methods, e.g., the adversarial loss in ADDA Tzeng et al. (2017). We consistently observe that this stabilizes the training and promotes faster convergence.

**Digit Classification.** We perform experiments on three public digits datasets: MNIST, USPS, and Street View House Numbers (SVHN), with three transfer tasks: USPS to MNIST (U $\rightarrow$ M), MNIST to USPS (M $\rightarrow$ U), and SVHN to MNIST (S $\rightarrow$ M). Classification accuracy on the target domain test set samples is adopted as the metric for measuring the performance. We perform domain-adaptive domain decomposition on state-of-the-art methods DANN

Table 2: Accuracy (%) on Digits for unsupervised domain adaptation.

| Methods | M $\rightarrow$ U | U $\rightarrow$ M | S $\rightarrow$ M | Avg. |
|---|---|---|---|---|
| DANN | - | - | 73.9 | |
| ADDA | 89.4 | 90.1 | 76.0 | 85.1 |
| CDAN+E | 95.6 | 98.0 | 89.2 | 94.3 |
| DANN + DAFD | 92.0 | 95.2 | 82.1 (**11.1**% ↑) | 89.8 |
| ADDA + DAFD | 91.4 | 94.8 | 82.9 | 89.7 (**5.5**% ↑) |
| CDAN+E + DAFD | 96.8 | 98.8 | 96.6 | 97.4 (**3.2**% ↑) |

Ganin et al. (2016), ADDA Tzeng et al. (2017), and CDAN+E Long et al. (2018). Quantitative comparisons are presented in Table 2.

**Office-31.** Office-31 is one of the most widely used datasets for visual domain adaptation, which has 4,652 images and 31 categories collected from three distinct domains: Amazon (A), Webcam (W), and DSLR (D). We evaluate all methods on six transfer tasks A → W, D → W, W → D, A → D, D → A, and W → A. Two feature extractors, AlexNet Krizhevsky et al. (2012) and ResNet He et al. (2016) are adopted for fair comparisons with underlying methods. Specifically, ImageNet initialization are widely used for ResNet in the experiments with Office-31, and we consistently observe that initialization is important for the training on Office-31. Therefore, when training ResNet based networks with domain-adaptive filter decomposition, we initialize the feature extractor using parameters decomposed from ImageNet initialization. The quantitative comparisons are in Table 3.

Table 3: Accuracy (%) on Office-31 for unsupervised domain adaptation (AlexNet and ResNet).

| | Method | A → W | D → W | W → D | A → D | D → A | W → A | Avg. |
|---|---|---|---|---|---|---|---|---|
| AlexNet | AlexNet (no adaptation) | 61.6±0.5 | 95.4±0.3 | 99.0±0.2 | 63.8±0.5 | 51.1±0.6 | 49.8±0.4 | 70.1 |
| | DANN Ganin et al. (2016) | 73.0±0.5 | 96.4±0.3 | 99.2±0.3 | 72.3±0.3 | 53.4±0.4 | 51.2±0.5 | 74.3 |
| | ADDA Tzeng et al. (2017) | 73.5±0.6 | 96.2±0.4 | 98.8±0.4 | 71.6±0.4 | 54.6±0.5 | 53.5±0.6 | 74.7 |
| | DANN + DAFD | 74.4±0.3 | 97.1±0.4 | 99.1±0.4 | 74.2±0.3 | 56.8±0.5 | 53.1±0.7 | 75.8 (**2.3%** ↑) |
| | ADDA + DAFD | 77.2±0.5 | 97.9±0.4 | 98.5±0.2 | 73.2±0.4 | 55.4±0.6 | 57.8±0.5 | 76.7 (**2.7%** ↑) |
| ResNet | ResNet-50 (no adaptation) | 68.4±0.2 | 96.7±0.1 | 99.3±0.1 | 68.9±0.2 | 62.5±0.3 | 60.7±0.3 | 76.1 |
| | DANN Ganin et al. (2016) | 82.0±0.4 | 96.9±0.2 | 99.1±0.1 | 79.7±0.4 | 68.2±0.4 | 67.4±0.5 | 82.2 |
| | ADDA Tzeng et al. (2017) | 86.2±0.5 | 96.2±0.3 | 98.4±0.3 | 77.8±0.3 | 69.5±0.4 | 68.9±0.5 | 82.9 |
| | CDAN+E Long et al. (2018) | 94.1±0.1 | 98.6±0.1 | 100.0±.0 | 92.9±0.2 | 71.0±0.3 | 69.3±0.3 | 87.7 |
| | DANN + DAFD | 86.4±0.4 | 96.8±0.2 | 99.2±0.1 | 84.4±0.4 | 70.5±0.4 | 68.8±0.4 | 84.35 (**2.3%** ↑) |
| | ADDA + DAFD | 86.8±0.4 | 97.7±0.1 | 98.4±0.1 | 80.5±0.3 | 71.1±0.4 | 69.1±0.5 | 83.9 (**1.2%** ↑) |
| | CDAN+E + DAFD | 95.6±0.1 | 98.8±0.1 | 100.0±0.0 | 93.5±0.2 | 76.6±0.5 | 71.3±0.4 | 89.3 (**1.8%** ↑) |

**Image Segmentation.** Beyond image classification tasks, we perform a challenging experiment on image segmentation to demonstrate the generality of the proposed domain-adaptive filter decomposition. We perform unsupervised adaptation from the GTA dataset Richter et al. (2016) (images generated from video games) to the Cityscapes dataset Cordts et al. (2016) (real-world images), which has a significant practical value considering the expensive cost on collecting annotations for image segmentation in real-world scenarios. Two underlying methods FCNs in the wild Hoffman et al. (2016) and AdaptSegNet Tsai et al. (2018) are adopted for comprehensive comparisons. Based on the underlying methods, all the convolutional layers are decomposed using domain-adaptive filter decomposition, and all the transpose-convolutional layers are kept sharing by both domains. For quantitative results in Table 4, we use intersection-over-union, i.e., IoU = $\frac{\text{TP}}{\text{TP}+\text{FP}+\text{FN}}$, where TP, FP, and FN are the numbers of true positive, false positive, and false negative pixels, respectively, as the evaluation metric. As with the previous examples, our method improves all state-of-the-art architectures. Qualitative results are shown in Figure A.2, and data samples are in Section C.1.

Table 4: Unsupervised DA for semantic segmentation: GTA → Cityscapes

| Methods | IoU | Class-wide IoU | | | | | | | | | | | | | | | | | | | |
|---|---|---|---|---|---|---|---|---|---|---|---|---|---|---|---|---|---|---|---|---|---|
| | | road | sidewalk | building | wall | fence | pole | t-light | t-sign | veg | terrain | sky | person | rider | car | truck | bus | train | mbike | bicycle |
| No Adapt (VGG) | 17.9 | 26.0 | 14.9 | 65.1 | 5.5 | 12.9 | 8.9 | 6.0 | 2.5 | 70.0 | 2.9 | 47.0 | 24.5 | 0.0 | 40.0 | 12.1 | 1.5 | 0.0 | 0.0 | 0.0 |
| No Adapt (ResNet) | 36.6 | 75.8 | 16.8 | 77.2 | 12.5 | 21.0 | 25.5 | 30.1 | 20.1 | 81.3 | 24.6 | 70.3 | 53.8 | 26.4 | 49.9 | 17.2 | 25.9 | 6.5 | 25.3 | 36.0 |
| FCN WLD (VGG) | 27.1 | 70.4 | 32.4 | 62.1 | 14.9 | 5.4 | 10.9 | 14.2 | 2.7 | 79.2 | 21.3 | 64.6 | 44.1 | 4.2 | 70.4 | 8.0 | 7.3 | 0.0 | 3.5 | 0.0 |
| AdaptSegNet (VGG) | 35.0 | 87.3 | 29.8 | 78.6 | 21.1 | 18.2 | 22.5 | 21.5 | 11.0 | 79.7 | 29.6 | 71.3 | 46.8 | 6.5 | 80.1 | 23.0 | 26.9 | 0.0 | 10.6 | 0.3 |
| AdaptSegNet (ResNet) | 42.4 | 86.5 | 36.0 | 79.9 | 23.4 | 23.3 | 23.9 | 35.2 | 14.8 | 83.4 | 33.3 | 75.6 | 58.5 | 27.6 | 73.7 | 32.5 | 35.4 | 3.9 | 30.1 | 28.1 |
| FCN WLD + DAFD | 32.7 (**20.7%** ↑) | 76.4 | 36.7 | 68.8 | 17.6 | 5.8 | 11.1 | 13.9 | 2.9 | 80.0 | 24.4 | 69.1 | 47.5 | 4.3 | 74.4 | 14.1 | 6.3 | 0.0 | 2.1 | 0.0 |
| AdaptSegNet (VGG) + DAFD | 36.4 (**4.0%** ↑) | 86.7 | 35.3 | 78.8 | 22.8 | 14.5 | 23.9 | 21.9 | 18.2 | 82.1 | 32.2 | 66.8 | 49.6 | 10.1 | 81.2 | 19.6 | 27.1 | 1.1 | 11.4 | 4.2 |
| AdaptSegNet (ResNet) + DAFD | 45.0 (**6.1%** ↑) | 88.2 | 38.5 | 8.12 | 25.0 | 23.8 | 22.9 | 35.1 | 14.4 | 84.9 | 34.1 | 79.9 | 59.5 | 29.1 | 75.5 | 30.1 | 35.2 | 2.9 | 28.7 | 29.1 |

## 6 CONCLUSION

We proposed to perform domain-invariant learning through domain-adaptive filter decomposition. To model domain shifts, convolutional filters in a deep convolutional network are decomposed over domain-adaptive bases to counter domain shifts, and cross-domain basis coefficients are constrained to unify common semantics. We present the intuitions of countering domain shifts by adapting bases through toy examples, and further provide theoretical analysis. Extensive experiments on multiple tasks validate that, by stacking domain-adaptive branched layers with filter decomposition, complex domain shifts in real-world scenarios can be bridged to produce domain-invariant representation, which are reflected by both experimental results and feature space visualizations, all this at virtual no additional memory or computational cost when adding domains.

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

## A    TOY EXPERIMENT

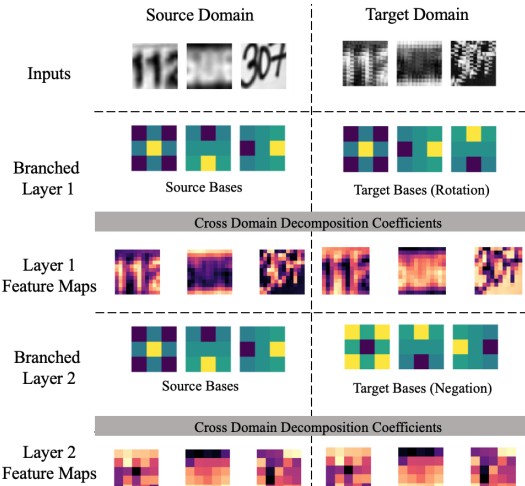

Figure A.1: Visualization of the toy example. The two columns visualize the inputs, features, and bases of the source domain and the target domain, respectively. Only the output feature in the first channel of each convolutional layer is visualized for comparison. Domain invariant features, the last row, are obtained by manually adapting source domain bases to generated target domain bases.

## B    SUPERVISED CROSS-DOMAIN FACE RECOGNITION

Besides standard domain adaptation, the proposed domain-adaptive filter decomposition can be extended to general tasks that involves more than one visual domain. Here we demonstrate this by performing experiments on supervised cross-domain face recognition. We adopt the NIR-VIS 2.0, which consists of 17,580 NIR (near infrared) and VIS (visible light) face images of 725 subjects, and perform cross-domain face recognition. We adopt VGG16 as the base network structure, branch all the convolutional layers with the proposed domain-adaptive filter decomposition, and train the network from scratch. In each convolutional layer, two bases are trained for modeling the NIR and the VIS domain, respectively. Specifically, one VIS image and one NIR image are fed simultaneously to the network, and the feature vectors of both domains are averaged to produce the final cross-domain feature, which is further fed into a linear classifier for classifying the identity. While the training is conducted using both domains simultaneously, we test the network under three settings including feeding single domain inputs only (VIS Acc and NIR Acc in Table 3(b)) and both domain inputs (VIS+NIR Acc in Table 3(b)). Quantitative comparisons demonstrate that branching with the proposed DAFD performs superiorly even with a missing input domain.

## C    DATASET SAMPLES AND QUALITATIVE RESULTS

### C.1    UNSUPERVISED DA FOR IMAGE SEGMENTATION

For the segmentation experiments in Section 5.5, we provide more qualitative results in Figure A.2, and some dataset samples in Figure A.3.

## D    COMPUTATION AND PARAMETERS

In Table A.1, we provide comparisons on additional parameters and computation introduced by one extra domain with and without the proposed domain-adaptive filter decomposition. The comparison reveals that domain-adaptive filter decomposition not only delivers superior performances but also saves both parameters and computation significantly.

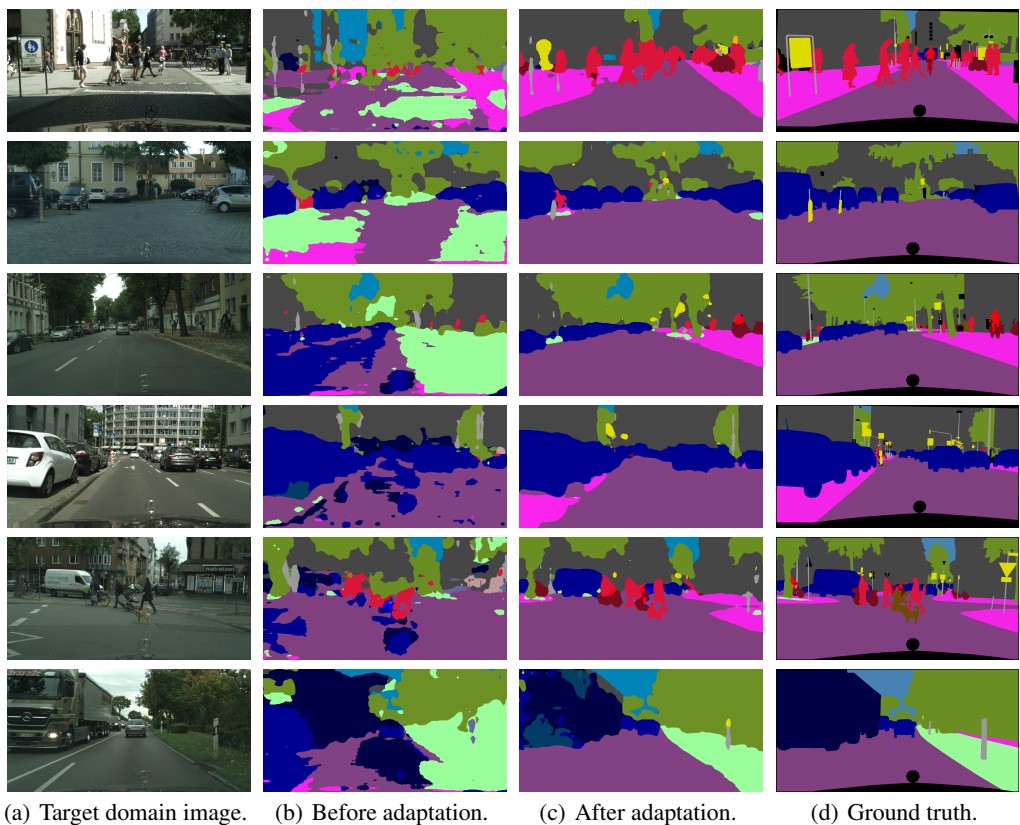

(a) Target domain image.   (b) Before adaptation.   (c) After adaptation.   (d) Ground truth.

Figure A.2: Qualitative results for domain adaptation segmentation. The samples are randomly selected from the validation subsets of Cityscapes.

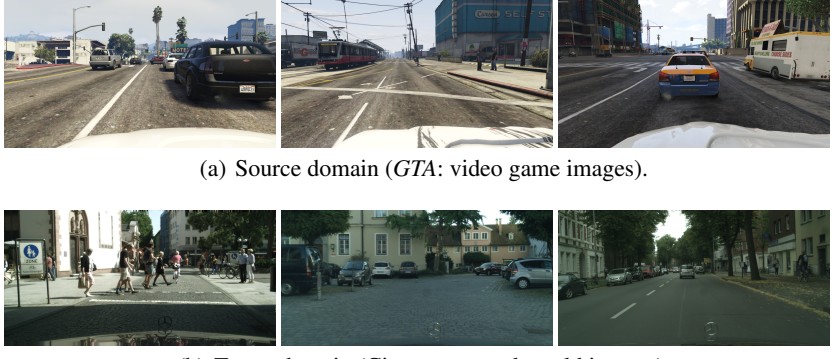

(a) Source domain (*GTA*: video game images).

(b) Target domain (Cityscapes: real-world images).

Figure A.3: Dataset samples for segmentation experiments (video games → street views).

# E   PROOFS IN SECTION 4

*Proof of Lemma 1.* We establish a few facts:

Fact 1. $|\nabla\tau|_\infty < \frac{1}{5}$ guarantees that, $\rho := I_d - \tau$,

$$||J\rho| - 1|, ||J\rho^{-1}| - 1| \leq 4|\nabla\tau|_\infty, \tag{A.1}$$

where $Jf = \det(\nabla f)$ denotes the determinate of the Jacobian matrix of the mapping $f : \mathbb{R}^2 \to \mathbb{R}^2$. The inequality can be verified by elementary calculation. When $\rho$ is a rigid motion then the r.h.s of equation A.1 is zero.

Table A.1: Comparisons on additional parameters and computation introduced by one extra domain. Comparisons are performed on VGG-16, with 6 basis elements and the input size of $224 \times 224$.

| Model | Regular VGG | VGG with DAFD |
|---|---|---|
| Parameters | 14.71M | 0.0007M |
| Flops | 15.38G | 10.75G |

**Fact 2.** $\rho$ is invertible, and odd symmetry of $\tau$ implies that $\rho$ and thus $\rho^{-1}$ are odd, namely $-\rho^{-1}(-u) = \rho^{-1}(u)$.

Define

$$y_1(u) := \sigma_b(x * D_\tau w) * f(u)$$

$$= \int_{\mathbb{R}^2} \sigma_b \left( \int_{\mathbb{R}^2} x(u + v - z)w(\rho(z))dz \right) f(-v)dv$$

$$= \int_{\mathbb{R}^2} \sigma_b \left( \int_{\mathbb{R}^2} x(u + v - \rho^{-1}(\tilde{z}))w(\tilde{z})|J\rho^{-1}(\tilde{z})|d\tilde{z} \right) f(-v)dv$$

and

$$\hat{y}_1(u) := \int_{\mathbb{R}^2} \sigma_b \left( \int_{\mathbb{R}^2} x(u + v - \rho^{-1}(\tilde{z}))w(\tilde{z})d\tilde{z} \right) f(-v)dv.$$

We have that

$$|y_1(u) - \hat{y}_1(u)| \le \int_{\mathbb{R}^2} \int_{\mathbb{R}^2} |x(u + v - \rho^{-1}(\tilde{z}))||w(\tilde{z})| \left| |J\rho^{-1}| - 1 \right| |f(-v)|d\tilde{z}dv \quad \text{(by } \sigma_b \text{ non-expansive)}$$

$$\le 4|\nabla\tau|_\infty \int_{\mathbb{R}^2} \int_{\mathbb{R}^2} |x(u + v - \rho^{-1}(\tilde{z}))||w(\tilde{z})||f(-v)|d\tilde{z}dv \quad \text{(by Fact 1)}$$

and thus

$$\|y_1 - \hat{y}_1\|_1 \le 4|\nabla\tau|_\infty \|x\|_1 \|w\|_1 \|f\|_1. \tag{A.2}$$

When $\rho$ is a rigid motion, $y_1 = \hat{y}_1$.

Also, let

$$y_2(u) := \sigma_b(x * w) * D_\tau^{-1} f(u)$$

$$= \int_{\mathbb{R}^2} \sigma_b \left( \int_{\mathbb{R}^2} x(u + v - z)w(z)dz \right) f(-\rho^{-1}(v))dv \quad \text{(by Fact 2)}$$

$$= \int_{\mathbb{R}^2} \sigma_b \left( \int_{\mathbb{R}^2} x(u + \rho(\tilde{v}) - z)w(z)dz \right) f(-\tilde{v})|J\rho(\tilde{v})|d\tilde{v}$$

and

$$\hat{y}_2(u) := \int_{\mathbb{R}^2} \sigma_b \left( \int_{\mathbb{R}^2} x(u + \rho(\tilde{v}) - z)w(z)dz \right) f(-\tilde{v})d\tilde{v}.$$

Similar to the proof of equation A.2, one can verify that

$$\|y_2 - \hat{y}_2\|_1 \le 4|\nabla\tau|_\infty \|x\|_1 \|w\|_1 \|f\|_1, \tag{A.3}$$

and the bound is zero when $\rho$ is a rigid motion.

It remains to bound $\|\hat{y}_1 - \hat{y}_2\|_1$. Note that by $\sigma_b$ being non-expansive again

$$|\hat{y}_1(u) - \hat{y}_2(u)| \le \int_{\mathbb{R}^2} \int_{\mathbb{R}^2} |x(u + v - \rho^{-1}(z)) - x(u + \rho(v) - z)||w(z)|dz|f(-v)|dv. \tag{A.4}$$

We claim that

$$\int_{\mathbb{R}^2} |x(u + v - \rho^{-1}(z)) - x(u + \rho(v) - z)|du \le |\nabla\tau|_\infty 2(2^{j_w} + 2^{j_f})\|\nabla x\|_1 \tag{A.5}$$

uniformly for $v$ and $z$. If true, with equation A.4 it gives that

$$\int_{\mathbb{R}^2} |\hat{y}_1(u) - \hat{y}_2(u)|du \le |\nabla\tau|_\infty 2(2^{j_w} + 2^{j_f})\|\nabla x\|_1 \|w\|_1 \|f\|_1$$

which proves the lemma together with equation A.2 and equation A.3.

Proof of equation A.5: We verify that for any fixed $v$, $z$,

$$\int_{\mathbb{R}^2} |x(u + v - \rho^{-1}(z)) - x(u + \rho(v) - z)| du \le \|\nabla x\|_1 |\nabla \tau|_\infty |v - \rho^{-1}(z)|, \tag{A.6}$$

by a direct calculation:

$$\begin{aligned}
(\text{l.h.s}) &\le \|\nabla x\|_1 |(v - \rho^{-1}(z)) - (\rho(v) - z)| \\
&= \|\nabla x\|_1 |\tau(v) - \tau(\rho^{-1}(z))| \\
&\le \|\nabla x\|_1 |\nabla \tau|_\infty |v - \rho^{-1}(z)|.
\end{aligned}$$

Then, combined with that $v \in 2^{j_f} B$ thus $|v| \le 2^{j_f}$, and $z \in 2^{j_w} B$ and thus $|\rho^{-1}(z)| \le \frac{1}{1-|\nabla \tau|_\infty} 2^{j_w} \le 2 2^{j_w}$ ($\tau(0) = 0$ by that $\tau$ is odd, and then $|\tau(\rho^{-1}(z))| \le |\nabla \tau|_\infty |\rho^{-1}(z)|$), the r.h.s of equation A.6 $\le 2(2^{j_w} + 2^{j_f})|\nabla \tau|_\infty \|\nabla x\|_1$, which proves equation A.5. $\qquad\square$

*Proof of Theorem 1.* We need a slightly generalized form of Lemma 1, which inserts multiple plain convolutional layers between $*w$ and $*f$, presented in Lemma 2.

Under the setting of the theorem, in the generative CNNs,

$$X_s = \sigma(\cdots \sigma(h * w_s^{(-L)} + b_s^{(-L)}) \cdots * w_s^{(-1)} + b_s^{(-1)}) \tag{A.7}$$

$$X_t = \sigma(\cdots \sigma(h * w_t^{(-L)} + b_t^{(-L)}) \cdots * w_t^{(-1)} + b_t^{(-1)}) \tag{A.8}$$

where $w_t^{(l)}$ and $b_t^{(l)}$ are defined by, $l = -L, \cdots, -1$,

$$w_t^{(l)} = D_l w_s^{(l)}, \quad \tilde{x}_0^{(l)} * w_t^{(l)} + b_t^{(l)} = \tilde{x}_0^{(l)} * w_s^{(l)} + b_s^{(l)}. \tag{A.9}$$

The notation $\tilde{x}^{(l)}$ stands for the $l$-th layer output in the target net from the input in the bottom $((-L)$-th) layer as $\tilde{x}^{(-L)} = h$, $\tilde{x}^{(0)} = X_t$, and $\tilde{x}_0^{(l)}$ for that from zero input in the bottom. In the feature CNNs, the $L$-th layer outputs are

$$F_s = \sigma(\cdots \sigma(X_s * w_s^{(1)} + b_s^{(1)}) \cdots * w_s^{(L)} + b_s^{(L)}) \tag{A.10}$$

$$F_t = \sigma(\cdots \sigma(X_t * w_t^{(1)} + b_t^{(1)}) \cdots * w_t^{(L)} + b_t^{(L)}) \tag{A.11}$$

where for $l = 1, \cdots, L$,

$$w_t^{(l)} = D_l w_s^{(l)}, \quad b_t^{(l)} = b_s^{(l)}.$$

The proof is by applying Lemma 2 recursively to the pair of layers indexed by $l$ and $-l$, from $l = 1$ to $L$. Denote $w_s^{(l)}$ by $w^{(l)}$, then $w_t^{(l)} = D_l w^{(l)}$, where $D_{-l} = D_l = D_{\tau_l}$, $l = 1, \cdots, L$. We also denote $b_s^{(l)}$ by $b^{(l)}$ and keep notation $b_t^{(l)}$ for negative $l$.

First, $l = 1$, in the target net,

$$\tilde{x}^{(1)} := \sigma(\sigma(\tilde{x}^{(-1)} * D_1 w^{(-1)} + b_t^{(-1)}) * D_1 w^{(1)} + b^{(1)})$$

Use the centering $\tilde{x}_c^{(-1)} := \tilde{x}^{(-1)} - \tilde{x}_0^{(-1)}$, it can be written as

$$\tilde{x}^{(1)} = \sigma(\sigma(\tilde{x}_c^{(-1)} * D_1 w^{(-1)} + \tilde{x}_0^{(-1)} * D_1 w^{(-1)} + b_t^{(-1)}) * D_1 w^{(1)} + b^{(1)}) \tag{A.12}$$

$$= \sigma(\sigma(\tilde{x}_c^{(-1)} * D_1 w^{(-1)} + (\tilde{x}_0^{(-1)} * w^{(-1)} + b^{(-1)})) * D_1 w^{(1)} + b^{(1)}) \text{ (by equation A.9)} \tag{A.13}$$

Applying Lemma 2 (or Lemma 1 for this case), taking $\tilde{x}_0^{(-1)} * w^{(-1)} + b^{(-1)}$ as the effective "$b$", we have that (using the non-expansiveness of $\sigma$ to take $r$ outside the last $\sigma$)

$$\tilde{x}^{(1)} = \sigma(\sigma(\tilde{x}_c^{(-1)} * w^{(-1)} + \tilde{x}_0^{(-1)} * w^{(-1)} + b^{(-1)}) * w^{(1)} + b^{(1)}) + r^{(1)} \tag{A.14}$$

$$= \sigma(\sigma(\tilde{x}^{(-1)} * w^{(-1)} + b^{(-1)}) * w^{(1)} + b^{(1)}) + r^{(1)} \tag{A.15}$$

$$:= \hat{x}^{(1)} + r^{(1)} \tag{A.16}$$

where, since $w^{(-1)}$, $w^{(1)}$ are supported on $2^{j_1} B$,

$$\|r^{(1)}\|_1 \le 4\varepsilon \left\{ 2^{j_1} \|\nabla \tilde{x}_c^{(-1)}\|_1 + 2\|\tilde{x}_c^{(-1)}\|_1 \right\}. \tag{A.17}$$

Next,

$$\tilde{x}^{(2)} := \sigma(\tilde{x}^{(1)} * D_2 w^{(2)} + b^{(2)}) \tag{A.18}$$

$$= \sigma((\hat{x}^{(1)} + r^{(1)}) * D_2 w^{(2)} + b^{(2)}) \text{ (by equation A.16)} \tag{A.19}$$

$$= \sigma(\hat{x}^{(1)} * D_2 w^{(2)} + b^{(2)}) + r^{(1)'} \tag{A.20}$$

where $\|r^{(1)'}\|_1 \leq \|r^{(1)}\|_1$ and observe the same bound as equation A.17, since neither $*w_t^{(2)}$ (Lemma 3(i)) nor applying $\sigma$ with bias expands the 1-norm. Using the brief notation $\sigma_l$ to denote the non-linear mapping with biases $b^{(l)}$, consider

$$\sigma_2(\hat{x}^{(1)} * D_2 w^{(2)}) = \sigma_2(\sigma_1(\sigma_{-1}(\tilde{x}^{(-1)} * w^{(-1)}) * w^{(1)}) * D_2 w^{(2)})$$
$$= \sigma_2(\sigma_1(\sigma_{-1}(\sigma(\tilde{x}^{(-2)} * D_2 w^{(-2)} + b_t^{(-2)}) * w^{(-1)}) * w^{(1)}) * D_2 w^{(2)})$$
$$= \sigma_2(\sigma_1(\sigma_{-1}(\sigma(\tilde{x}_c^{(-2)} * D_2 w^{(-2)} + \tilde{x}_0^{(-2)} * w^{(-2)} + b^{(-2)})$$
$$* w^{(-1)}) * w^{(1)}) * D_2 w^{(2)}), \quad \text{(by equation A.9)}$$

by Lemma 2, it equals (using the non-expansiveness of $\sigma_2$ to take $r^{(2)}$ outside)

$$\sigma_2(\sigma_1(\sigma_{-1}(\sigma(\tilde{x}_c^{(-2)} * w^{(-2)} + \tilde{x}_0^{(-2)} * w^{(-2)} + b^{(-2)}) * w^{(-1)}) * w^{(1)}) * w^{(2)}) + r^{(2)}$$
$$= \sigma_2(\sigma_1(\sigma_{-1}(\sigma(\tilde{x}^{(-2)} * w^{(-2)} + b^{(-2)}) * w^{(-1)}) * w^{(1)}) * w^{(2)}) + r^{(2)}$$
$$:= \hat{x}^{(2)} + r^{(2)}$$

where

$$\|r^{(2)}\|_1 \leq 4\varepsilon \left\{ 2^{j_2} \|\nabla \tilde{x}_c^{(-2)}\|_1 + 2\|\tilde{x}_c^{(-2)}\|_1 \right\}. \tag{A.21}$$

Inserting back to equation A.20,

$$\tilde{x}^{(2)} = \hat{x}^{(2)} + r^{(1)'} + r^{(2)}$$

thus $\|\tilde{x}^{(2)} - \hat{x}^{(2)}\|_1$ is bounded by the sum of equation A.17 and equation A.21.

Continue the process, $\hat{x}^{(l)}$ denotes the $l$-th layer output in the source CNN (after $l$ times correction in the target CNN) by feeding $\tilde{x}^{(-l-1)}$ from the $(-l)$-th layer, where $\tilde{x}^{(-l-1)}$ is the output in the (un-corrected) generative target CNN after the first $(L-l)$ layers. By that $\tilde{x}^{(-L)} = x^{(-L)} = h$, and that $F_t = \tilde{x}^{(L)}$, $F_s = x^{(L)}$, repeating the argument $L$ times gives that

$$\|F_s - F_t\|_1 \leq 4\varepsilon \sum_{l=1}^{L} (2^{j_l} \|\nabla \tilde{x}_c^{(-l)}\|_1 + 2\|\tilde{x}_c^{(-l)}\|_1),$$

and when $(I_d - \rho_l)$ are rigid motions, the 2nd term for each $l$ vanishes.

We claim that

Claim 3. For $l = -L, \cdots, -1$, $\|\nabla \tilde{x}_c^{(l)}\|_1 \leq \|\nabla h\|_1$, and $\|\tilde{x}_c^{(l)}\|_1 \leq \|h\|_1$.

which suffices to prove the theorem.

Proof of Claim 3: No that in the bottom layer $\tilde{x}_c^{(-L)} = \tilde{x}^{(-L)} = h$. For $l = -L + 1, \cdots, -1$,

$$\|\tilde{x}_c^{(l)}\|_1 = \|\tilde{x}^{(l)} - \tilde{x}_0^{(l)}\|_1$$
$$= \|\sigma_l(\tilde{x}^{(l-1)} * w_t^{(l-1)}) - \sigma_l(\tilde{x}_0^{(l)} * w_t^{(l-1)})\|_1$$
$$\leq \|\tilde{x}^{(l-1)} * w_t^{(l-1)} - \tilde{x}_0^{(l)} * w_t^{(l-1)}\|_1 \quad \text{(by that } \sigma_l \text{ non-expansive)}$$
$$\leq \|\tilde{x}^{(l-1)} - \tilde{x}_0^{(l)}\|_1 \quad \text{(by that } \|w_t^{(l-1)}\|_1 \leq 1 \text{ and Lemma 3(i))}$$
$$= \|\tilde{x}_c^{(l-1)}\|_1.$$

Recursing the inequality gives that $\|\tilde{x}_c^{(l)}\|_1 \leq \|h\|_1$. Similarly,

$$\|\nabla \tilde{x}_c^{(l)}\|_1 = \|\nabla \tilde{x}^{(l)}\|_1 = \text{TV}[\sigma_l(\tilde{x}^{(l-1)} * w_t^{(l-1)})]$$
$$\leq \text{TV}[\tilde{x}^{(l-1)} * w_t^{(l-1)}] \quad \text{(by that } \sigma_l \text{ does not increase total variation)}$$
$$= \|\nabla(\tilde{x}^{(l-1)} * w_t^{(l-1)})\|_1$$
$$\leq \|\nabla \tilde{x}^{(l-1)}\|_1 = \|\nabla \tilde{x}_c^{(l-1)}\|_1, \quad \text{(by that } \|w_t^{(l-1)}\|_1 \leq 1 \text{ and Lemma 3(ii))}$$

and thus $\|\nabla \tilde{x}_c^{(l)}\|_1 \leq \|\nabla h\|_1$. This proves Claim 3. $\qquad \square$

**Lemma 2.** *Suppose filters $w$, $f_1, \cdots, f_m$, $f$ satisfy that the 1-norm are all bounded by 1, and $w$ and $f$ are supported on $2^j B$. The sequence of $\sigma_l$, denoting non-linear function with bias, for $l = 0, \cdots, m$ are non-expansive. $D_\tau$ is a spatial transform where $\tau$ is odd and $|\nabla \tau|_\infty \leq \varepsilon < \frac{1}{5}$. Then*

$$\sigma_m(\cdots \sigma_1(\sigma_0(x * D_\tau w) * f_1) \cdots * f_m) * f$$

*approximates*

$$\sigma_m(\cdots\sigma_1(\sigma_0(x*w)*f_1)\cdots*f_m)*D_\tau^{-1}f$$

*up to an error whose 1-norm is bounded by*

$$4\varepsilon\left\{2^j\|\nabla x\|_1+2\|x\|_1\right\},$$

*and the second term vanishes if $(I_d-\tau)$ is a rigid motion.*

*Proof of Lemma 2.* The proof uses the same technique as in the proof of Lemma 1. Omitting subscript $\mathbb{R}^2$ in the integral, let

$$y_1(u)=\int\sigma_m(\int\cdots\sigma_1(\int\sigma_0(\int x(u+v_1+\cdots+v_m+v-\rho^{-1}(z))w(z)|J\rho^{-1}|dz)$$

$$f(-v_1)dv_1)\cdots f_m(-v_m)dv_m)f(-v)dv,$$

$$\hat{y}_1(u)=\int\sigma_m(\int\cdots\sigma_1(\int\sigma_0(\int x(u+v_1+\cdots+v_m+v-\rho^{-1}(z))w(z)dz)$$

$$f(-v_1)dv_1)\cdots f_m(-v_m)dv_m)f(-v)dv.$$

By Fact 1, that $\sigma_j$ are all non-expansive and that the 1-norm of all the filters are bounded by 1,

$$\int|y_1(u)-\hat{y}_1(u)|du\le4\varepsilon\|x\|_1.$$

Also,

$$y_2(u)=\int\sigma_m(\int\cdots\sigma_1(\int\sigma_0(\int x(u+v_1+\cdots+v_m+\rho(v)-z)w(z)dz)$$

$$f(-v_1)dv_1)\cdots f_m(-v_m)dv_m)f(-v)|J\rho|dv,$$

$$\hat{y}_2(u)=\int\sigma_m(\int\cdots\sigma_1(\int\sigma_0(\int x(u+v_1+\cdots+v_m+\rho(v)-z)w(z)dz)$$

$$f(-v_1)dv_1)\cdots f_m(-v_m)dv_m)f(-v)dv.$$

Similarly,

$$\int|y_2(u)-\hat{y}_2(u)|du\le4\varepsilon\|x\|_1.$$

Same as before, with $\rho$ being a rigid motion, $\|y_1-\hat{y}_1\|$ and $\|y_2-\hat{y}_2\|$ are both zero.

It remains to bound $\|\hat{y}_1-\hat{y}_2\|_1$. Observe that

$$\int|\hat{y}_1(u)-\hat{y}_2(u)|du\le\int\cdots\int dv|f(-v)|dv_m|f(-v_m)|\cdots dv_1|f(-v_1)|dz|w(z)|$$

$$\int du|x(u+v_1+\cdots+v_m+v-\rho^{-1}(z))-x(u+v_1+\cdots+v_m+\rho(v)-z)|,\quad\text{(A.22)}$$

and similarly as in proving Lemma 1, one can verify that for any fixed $v_1,\cdots,v_m,v,z$,

$$\int|x(u+v_1+\cdots+v_m+v-\rho^{-1}(z))-x(u+v_1+\cdots+v_m+\rho(v)-z)|du$$

$$\le\|\nabla x\|_1|\nabla\tau|_\infty|v-\rho^{-1}(z)|\le\varepsilon2(2^j+2^j)\|\nabla x\|_1.$$

Inserting back to equation A.22, and again by that the 1-norm of all the filters are bounded by 1, we have that $\|\hat{y}_1-\hat{y}_2\|_1\le4\varepsilon2^j\|\nabla x\|_1$. $\square$

**Lemma 3.** *Let $x$ and $w$ be smooth and compactly supported on $\mathbb{R}^2$, then*

*(i) $\|x*w\|_1\le\|x\|_1\|w\|_1$.*

*(ii) $\|\nabla(x*w)\|_1\le\|\nabla x\|_1\|w\|_1$.*

*Proof of Lemma 3.* For (i),

$$\|x*w\|_1=\int_{\mathbb{R}^2}|\int_{\mathbb{R}^2}x(u-v)w(v)dv|du\le\int_{\mathbb{R}^2}\int_{\mathbb{R}^2}|x(u-v)||w(v)|dudv=\|x\|_1\|w\|_1.$$

For (ii),

$$\|\nabla(x*w)\|_1=\int_{\mathbb{R}^2}|\nabla_u(\int_{\mathbb{R}^2}x(u-v)w(v)dv)|du$$

$$=\int_{\mathbb{R}^2}|\int_{\mathbb{R}^2}\nabla_ux(u-v)w(v)dv|du$$

$$\le\int_{\mathbb{R}^2}\int_{\mathbb{R}^2}|\nabla_ux(u-v)||w(v)|dudv$$

$$=\|\nabla x\|_1\|w\|_1.$$

$\square$

