# OpenReview forum: "Domain-invariant Learning using Adaptive Filter Decomposition"
_ICLR.cc/2020/Conference — Reject_

### Official Review · AnonReviewer3 · 2019-10-21
**Official Blind Review #3**

**Rating:** 6

**Review:**

The paper proposes an approach to learning domain invariant representations using the adaptive decomposition of the convolutional filters. The approach is similar to methods that use multi-stream networks (a stream for each domain), but using the filer decomposition scheme, the authors avoid the issue of excessive increase in the number of parameters typical in fully multi-stream architectures.

This is achieved by learning a separate basis for convolutional filters for each domain while sharing the basis coefficients across domains. This encourages shared semantics across domains while maintaining a balance between the network expressiveness and the computational complexity.

The authors argue that the basis learned for each domain can be understood as correction/alignment mappings to bring together the representations of each domain. A toy example presented is convincing and shows the correction basis learned in a simple synthetic case. Theoretical arguments are provided to show that the proposed correction scheme covers a large range of possible domain shifts.

The authors also show that plugging the decomposition scheme into existing CNN based unsupervised domain adaptation algorithms results in consistent improvements across methods and datasets.

How does the method compare to approaches that learn to adapt the representations using conditional/adaptive batch norm [1,2]?

Overall, the paper was well motivated and easy to read. The methods appear to be a useful addition to tools available for domain invariant learning.

[1] Chang, Woong-Gi, Tackgeun You, Seonguk Seo, Suha Kwak, and Bohyung Han. "Domain-Specific Batch Normalization for Unsupervised Domain Adaptation." In Proceedings of the IEEE Conference on Computer Vision and Pattern Recognition, pp. 7354-7362. 2019.
[2] Kumar, Abhishek, Prasanna Sattigeri, Kahini Wadhawan, Leonid Karlinsky, Rogerio Feris, Bill Freeman, and Gregory Wornell. "Co-regularized alignment for unsupervised domain adaptation." In Advances in Neural Information Processing Systems, pp. 9345-9356. 2018.



**Experience Assessment:**

I have published in this field for several years.

**Review Assessment: Checking Correctness Of Derivations And Theory:**

I assessed the sensibility of the derivations and theory.

**Review Assessment: Checking Correctness Of Experiments:**

I carefully checked the experiments.

**Review Assessment: Thoroughness In Paper Reading:**

I read the paper at least twice and used my best judgement in assessing the paper.

---

> ### Author Response · Authors · 2019-11-13
> **Thanks and response to Reviewer #3**
>
>
> Thank you for your support and the suggestions on important related works.
>
> Our method obtains comparable results to those reported in [1] for unsupervised domain adaptation. Since [1] utilizes traditional convolutional layers, as demonstrated in Tables 2,3,4, we believe our domain-adaptive filter decomposition can be plugged into methods in [1] for further performance improvement.
> For [2], it’s hard to directly compare due to the lack of shared experiments. And after reading the paper we believe our method is also compatible with the co-regularized alignment proposed in [2] to bring further improvement.
>
> Although the extensive experiments with very diverse datasets and state-of-the-art architectures already included in our manuscript indicate the generality of the proposed framework, we will validate our hypothesis by attempting to reproduce the methods in [1] [2], which is a bit more time consuming than the allowed in the rebuttal period due to the lack of their official implementations. And we will keep this as a direction of future efforts.

---

### Official Review · AnonReviewer2 · 2019-10-25
**Official Blind Review #2**

**Rating:** 1

**Review:**

This paper introduces a way to decompose features for better domain adaptation via learning domain-invariant representations. The main advantage of the proposed approach is that by only introducing a few model parameters, the proposed approach could quickly adapt to new domains. Similar idea has been proposed in [1], where the authors propose to use domain-specific encoders to extract domain-invariant features. Compared with [1], the main novelty of this paper lies in a new feature decomposition of the traditional convolution layer.

My main concern is that this paper seems to miss a significant line of recent work on learning domain-invariant representations [2-3]. Basically, it has been shown that invariant representations provably hurt generalization on the target domain when the marginal label distributions are different between the source and target domains. Note that such result also holds when different feature extractors are used in source and target domains, so the model proposed in this manuscript is also subject to such inherent tradeoffs.


[1].    Domain Separation Networks, NIPS 2016.
[2].    On Learning Invariant Representations for Domain Adaptation, ICML 2019.
[3].    Domain Adaptation with Asymmetrically-Relaxed Distribution Alignment, ICML 2019.


**Experience Assessment:**

I have published in this field for several years.

**Review Assessment: Checking Correctness Of Derivations And Theory:**

I assessed the sensibility of the derivations and theory.

**Review Assessment: Checking Correctness Of Experiments:**

I assessed the sensibility of the experiments.

**Review Assessment: Thoroughness In Paper Reading:**

I read the paper at least twice and used my best judgement in assessing the paper.

---

> ### Author Response · Authors · 2019-11-13
> **Thanks and response to Reviewer #2**
>
> Thanks for bringing several very recent and related references, which have been now included and discussed in the revised paper.
>
> * Novelty over [1].
>
> Instead of another invariant-learning method, the main message of our paper is to show that one extremely simple and computationally/memory efficient way to add invariant-learning to existing state-of-the-art CNN models is to represent each convolutional ﬁlter as a linear combination of a small set of basis elements. This is innovative in the sense that is not about a new architecture designed to outperform others for a given task and a given dataset, but about providing a plug-and-play framework, here exemplified with numerous architectures and examples and further supported by theory.
>
> In this way, each convolution layer is decomposed into two convolution layers, a basis layer followed by a coefficient layer. Since we have only spatial-wise convolution at a basis layer, and only channel-wise convolution at a coefficient layer, we now obtain a unique opportunity to “absorb” domain spatial variations if we make each basis layer domain-specific, but channel mixing coefficients shared across domains. All these are not feasible with traditional convolution layers without the proposed framework, where convolutions are performed simultaneously across space and channels.
>
> As demonstrated in Tables 2,3,4, by simply plugging the proposed domain ﬁlter decomposition into existing state-of-the-art CNN architectures, we always observe performance improvements in diverse tasks. This is thereby a framework and not a particular new architecture tailored to a task/dataset.
>
>
> * Tradeoff concern in [2,3].
>
> We find [2,3] do not conflict with our contributions, as detailed next.
>
> We believe neither the authors nor their empirical findings/theoretic arguments in [2] at all infer that efforts in invariant representation learning are futile; that is clearly not the spirit of the very important messages and contributions in [2]. For example, in our street-view image segmentation example, we perform unsupervised adaptation from video games images to real-world street-view images, which is obviously of extremely high practical value. Furthermore, the proposed framework allows to work with multiple domains simultaneously as illustrated in our paper. Both Table 4 and Figure A.2, for example, clearly indicate the success of the proposed invariant representation learning on providing invariant features. Such empirical success is further explained in the paper through a rigorous theoretical analysis of the provable invariance with domain-specific bases, under a large range of possible domain shifts (as common in the field, the theory lags behind the practical success, explaining only parts of it for the moment).
>
> We agree with [2] that there are situations in unsupervised DA where learning invariant representation and minimizing source domain loss may not guarantee generalization in the target domain. But such finding cannot deny the value of domain invariant feature learning, which is the focus of this paper, to domain adaptation tasks. On the contrary, as shown by the [4] and many other theoretical results, under proper settings, the learning of invariant features are the key for domain adaptation and learning of causal factors.
>
>
> [3] shares similar arguments with [2], and further proposes alignment methods that alleviate the negative effects introduced by different marginal label distributions. As demonstrated with extensive experiments, the proposed domain-adaptive filter decomposition can be plug-and-played in diverse architectures and diverse settings, both supervised and unsupervised, to improve over current baselines. With the improved training losses for unsupervised DA in [3], we can expect even further performance improvement by plugging our filter decomposition. We will validate our hypothesis by the time of the conference by attempting to reimplement losses in [3].
>
> In summary, we propose domain-adaptive filter decomposition as a plug-and-play method for various invariant feature learning tasks, including but not limited to unsupervised domain adaptation. And the theoretical result in our paper proves the alignment of deep representations at the level of individual image sample, without assuming any distribution of samples conditioned on classes. It is an image data representation analysis, not a probabilistic one. Thus our consideration by decomposing CNN is orthogonal to the issue raised by [2, 3].
>
> As mentioned before, we have now incorporated these important references and commented on it in the revised paper.
>
> [1]. Domain Separation Networks, NIPS 2016.
> [2]. On Learning Invariant Representations for Domain Adaptation, ICML 2019.
> [3]. Domain Adaptation with Asymmetrically-Relaxed Distribution Alignment, ICML 2019.
> [4]. Invariance, causality and robustness, arXiv preprint arXiv:1812.08233 (2018).

---

### Official Review · AnonReviewer1 · 2019-10-27
**Official Blind Review #1**

**Rating:** 6

**Review:**

Summarize what the paper claims to do/contribute.
* The paper proposes to perform domain adaptation via separating domain-specific and cross-domain features, by what is referred to as "domain-adaptive filter decomposition". Each domain contributes its own share of features to be combined by a subsequent common layer. The method was benchmarked against competing methods on simple classification tasks and a hard semantic segmentation task.

Clearly state your decision (accept or reject) with one or two key reasons for this choice.
Weak Accept.
* I think the paper was very well written, the explanations were clear and the technical contributions seem sound.
* The experiments were satisfying for the most part. I would have wanted to see MNIST->SVHN for the unsupervised case as well, as this is a particularly hard one.

Provide supporting arguments for the reasons for the decision.
* Please do not use the Office dataset, it is commonly used in unsupervised domain adaptation papers, especially older ones, but it's hard to tell anything about proposed methods from this dataset as there is label pollution and not enough samples per class to be used with neural nets.
* For GTA->Cityscapes you are missing a few works eg, the CYCADA work. Also please use citations in the tables if you did not yourselves run experiments (as to make it clear that experimental protocols also might be slightly different etc).

Provide additional feedback with the aim to improve the paper. Make it clear that these points are here to help, and not necessarily part of your decision assessment.
* Tables 1 & 2: It'd be better to not refer to methods as A1,2,3 but rahter with some specific names or explicitly describe what these abbreviations mean in the caption of the table.

**Experience Assessment:**

I have published in this field for several years.

**Review Assessment: Checking Correctness Of Derivations And Theory:**

I assessed the sensibility of the derivations and theory.

**Review Assessment: Checking Correctness Of Experiments:**

I carefully checked the experiments.

**Review Assessment: Thoroughness In Paper Reading:**

I made a quick assessment of this paper.

---

> ### Author Response · Authors · 2019-11-13
> **Thanks and response to Reviewer #1**
>
> Thank you for your support and all the insightful suggestions.
>
> * MNIST->SVHN
>
> We did an experiment on MNIST->SVHN with DANN. We only observe ~36% accuracy on the target domain using DANN alone, while DANN with the proposed domain-adaptive filter decomposition plugged in achieves ~48%.
>
> * GTA->Cityscapes comparison choices
>
> As mentioned at the beginning of Section 5.2 (and also stressed in related work), one main goal of our experiments is to show that by simply plugging the proposed domain-adaptive ﬁlter decomposition into existing state-of-the-art CNN architectures, we always observe performance improvements in various tasks. We opted them for diverse datasets and diverse architectures. This is critical to show the universal validity of the proposed framework.
>
> For the GTA->Cityscapes experiments in particular, we focus on methods that utilize shared network structure to learn invariant features for multiple domains, validating the effectiveness of simply plugging in domain decomposed filters. Methods such as CYCADA remap source training data into the target domain for domain adaptation and are thus not included for comparisons.
>
> All the experiments are conducted based on the official public implementations released by the authors with exactly the same settings as in the original papers.
>
> Thanks for the suggestions for the name scheme in Table 1. We have updated as suggested.

---

### Decision · Program_Chairs · 2019-12-19

**Decision:**

Reject

**Comment:**

The paper proposes a domain-adaptive filter decomposition method via separating domain-specific and cross-domain features, towards learning invariant representations for unsupervised domain adaptation.

Overall, this well-written paper is well motivated with a better technique for learning invariant representations using convolutional filters. Nonetheless, reviewers still have major concerns: 1) the novelty of the paper may be marginal given the significant line of recent work on learning domain-invariant representations; 2) when the label distributions differ, learning invariant representations can only lead to worse target generalizations; 3) the provided theory has an unclear connection to the presented filter decomposition method. The paper can be strengthened by further discussions on how to mitigate the aforementioned negative results.

Hence I recommend rejection.